# Patterns of Parental Reactions to Their Children’s Negative Emotions: A Cluster Analysis with a Clinical Sample

**DOI:** 10.3390/ijerph19116844

**Published:** 2022-06-03

**Authors:** Ana Isabel Pereira, Catarina Santos, Luísa Barros, Magda Sofia Roberto, Joana Rato, Ana Prata, Cristina Marques

**Affiliations:** 1CICPSI, Faculdade de Psicologia, Universidade de Lisboa, Alameda da Universidade, 1649-013 Lisboa, Portugal; lbarros@psicologia.ulisboa.pt (L.B.); msroberto@psicologia.ulisboa.pt (M.S.R.); 2Hospital D. Estefânia, Centro Hospitalar Lisboa Central, 1169-045 Lisboa, Portugal; ccsantos.gm@gmail.com (C.S.); anateresa.prata@chlc.min-saude.pt (A.P.); cristina.marq@hotmail.com (C.M.); 3Faculdade de Psicologia, Universidade de Lisboa, Alameda da Universidade, 1649-013 Lisboa, Portugal; joana14@campus.ul.pt

**Keywords:** child externalizing and internalizing problems, cluster analysis, parental reactions to children’s negative emotions, parent adjustment

## Abstract

Parents’ emotion socialization practices are an important source of influence in the development of children’s emotional competencies This study examined parental reactions to child negative emotions in a clinical sample using a cluster analysis approach and explored the associations between clusters of parents’ reactions and children’s and parents’ adjustment. The sample comprised 80 parents of Portuguese children (aged 3–13 years) attending a child and adolescent psychiatry unit. Measures to assess parental reactions to children’s negative emotions, parents’ psychopathological symptoms, parents’ emotion dysregulation, and children’s adjustment were administered to parents. Model-based cluster analysis resulted in three clusters: low unsupportive, high supportive, and inconsistent reactions clusters. These clusters differed significantly in terms of parents’ psychopathological symptoms, emotion dysregulation, and children’s adjustment. A pattern characterized by high supportive reactions to the child’s emotions was associated with higher levels of children’s adjustment. On the other hand, an inconsistent reactions pattern was associated with the worst indicators of children’s adjustment and parental emotion dysregulation. These results suggest the importance of supporting parents of children with emotional and behavioural problems so that they can be more responsive to their children’s emotional manifestations.

## 1. Introduction

There has been a substantial increase in research suggesting emotion regulation is a psychological dimension that is central to children’s development and health [1]. Children need to have good emotional regulation skills to overcome fundamental challenges and tasks in their daily life, such as playing with peers, initiating and maintaining friendships, succeeding in school activities, responding adequately to parents’ solicitations, overcoming frustration, and carrying out other daily activities in emotionally loaded situations. Childhood is a critical period for developing emotional competencies, and parents are considered the most important sources of influence in the development of these competencies, at least in early childhood [2].

Expanding previous work on the socialization of emotion by Eisenberg and colleagues [3,4], Morris et al. [1] proposed three key processes through which parents impact children’s emotion regulation: (a) observation (e.g., emotion contagion, modelling), (b) parenting practices (e.g., parents’ reaction to children’s emotions, emotional coaching), and (c) family emotional climate (e.g., parents’ emotional expressivity, parenting style). Several subsequent studies support this tripartite model of parents’ influence [1]. From these three processes, parenting practices have received particular attention, possibly because they are intentional and purposive parenting behaviours and easier to modify through interventions to promote children’s positive development.

Parenting emotion socialization practices occur in the context of other parental dimensions such as beliefs and socialization objectives related to emotion. Parental meta-emotion philosophy is defined as the feelings and thoughts that parents have concerning emotions [5]. Previous literature distinguishes two main meta-emotion philosophies: emotion coaching and emotion dismissing. Parents with an emotion coaching philosophy view emotions as opportunities for learning; they are aware of emotions, even those of low intensity, and use supportive emotion-related practices and reactions (e.g., emotion-focused reactions, encouraging emotion expression, and problem-focused reactions). On the other hand, parents with an emotion dismissing meta-emotion philosophy dismiss the importance of emotions; they convey to the child that emotions are unimportant or undesirable, and use unsupportive emotion-related practices and reactions (e.g., punishing, minimizing or ignoring the child’s emotions, or reacting with distress to the child’s emotions). Through these meta-emotion philosophies, parents might indirectly influence their child’s emotional self-regulation, both by being emotional role models and conveying values regarding the expression and control of emotions or by directly influencing by providing feedback and contingencies for the child’s emotional behaviour [6].

Since the work of Eisenberg [3] and Gottman [5], two decades of research support the relation between parents’ supportive and unsupportive emotion-related behaviours, particularly parental reactions to the child’s negative emotions, and children’s adjustment and development [7,8]. Parents’ unsupportive reactions to children’s negative emotions are associated with children’s psychopathology, externalizing [9,10] and internalizing problems [10,11], and lower levels of socio-emotional competence [12]. On the other hand, parents’ supportive reactions to children’s negative emotions seem to have a less clear pattern of associations [12], although some evidence suggests significant relations with positive aspects of children’s emotional and social functioning [10,13].

One important advantage of the meta-emotion philosophy approach to parents’ emotion socialization is its holistic focus on parents’ response to the child’s emotions. Gottman, Katz, and Hooven [5] presented a proposal of two main meta-emotion philosophies presupposing some degree of consistency in parents’ active and purposeful responses to their child’s emotion. Previous work using a dimensional approach supports this view, showing a considerable degree of consistency between parents’ responses to children’s emotions, with significant positive correlations within dimensions of supportive or negative practices, and a negative correlation between negative and positive practices [14]. Nevertheless, there is also evidence of inconsistency, especially in specific populations. A study conducted by Shadur and Hussong [15] observed that a clinical sample of substance-dependent mothers used higher levels of supportive and non-supportive reactions to children’s negative emotions than a community sample of mothers.

Empirical research in this domain has been mainly conducted using a variable-centred approach. However, parents do not react to children’s emotions with single responses; they use multiple and diversified strategies. A person-centred approach can be used to describe individual differences based on the grouping of the different reactions and responses used by the same parent [16]. This approach has higher ecological validity because it enables describing the different parental reactions that co-occur simultaneously in the natural environment and emphasizes the interaction effects between various target variables [17]. Furthermore, this approach allows the studying of inconsistency in parents’ reactions to children’s emotions.

To our knowledge, only two studies [16,17]—using community samples–adopted a person-centred approach to explore the emergence of patterns in parents’ responses to children’s emotions. The first study [18], conducted with 76 parents of young children (18 months–5 years), identified two clusters of parent emotion socialization behaviours, low-involvement parents (lower levels of positive and negative expressiveness, coaching, and dismissing behaviours) and high-involvement parents (higher levels of all emotion socialization dimensions). A second study [19], conducted with 51 families with school-aged children (8–12 years), found three patterns of reactions to the child’s sadness, supportive (high supportive and low non-supportive from both mother and father), not supportive (low supportive reactions from both mother and father), and father dominant (high supportive and non-supportive reactions from the father and low supportive and non-supportive reactions from the mother). The results of these studies suggest the presence of patterns characterized by some inconsistency in parents’ responses and a varied pattern of parents’ reactions, not limited to the two main meta-emotion philosophies.

The current study had four main objectives. The first objective was to characterize parental reactions to children’s negative emotions using a cluster analysis approach, and explore whether these profiles are characterized by consistent responses toward the child’s emotions. We aimed to identify how seven parental reactions to children’s negative emotions (i.e., punitive, distress, minimization, ignoring, problem-focused, expressive encouragement, emotion-focused) combine to form different parental emotion socialization profiles. Secondly, we examined how the child’s age is associated with different patterns of parental reactions to negative child emotions. Previous studies adopting a person-centred approach were conducted with parents of children with a more restricted age range, preventing this analysis. Given the exploratory nature of this objective and the scarcity of previous research, no hypothesis was formulated. Thirdly, we aimed to study the relationship between different types of parental reactions to the child’s negative emotions, parent psychopathological symptoms, and emotion dysregulation. Parent–child interactions can be emotionally challenging [20], and parents competent in regulating their emotions are better prepared to use supportive emotion strategies [21,22]. On the other hand, parents with emotion regulation problems are expected to be less well equipped to react in constructive ways to their child’s emotions and may react more negatively. Previous empirical work indicated significant associations between parental reactions to children’s negative emotions and parents’ mental health and emotion regulation [23]. We hypothesize that the different patterns of parents’ reactions should be differentially associated with parents’ psychopathological symptoms and parents’ emotion dysregulation.

Finally, the current study examined the associations between parental reactions to children’s negative emotions and the child’s internalizing and externalizing problems. As mentioned, several studies have shown significant associations between parental reactions to children’s negative emotions and children’s psychopathology symptoms [9]. However, to our knowledge, this is one of the first studies to adopt a typological approach to explore these relations. Miller-Slough et al. [18] found significant associations between different clusters of parents’ reactions and children’s symptomatology, but the results are difficult to generalize because the authors focused only on parents’ responses to children’s sadness. Nevertheless, based on the studies that used a dimensional approach, we hypothesize that a pattern characterized by more unsupportive and less supportive parental reactions would be associated with more externalizing and internalizing problems in children.

The current study also innovates in using a clinical sample. Most previous studies on parents’ reactions to children’s emotions were conducted with community samples. Research has associated a lack of emotional self-regulation with children’s mental health problems [24]. Children with mental health problems express negative emotionality more frequently and have more difficulties in successfully regulating these emotional displays [9]. Growing evidence pointing to emotion regulation deficits in the development of internalizing and externalizing problems indicates that emotion regulation might be a transdiagnostic factor underlying diverse mental health conditions [25]. Therefore, a novel contribution of this study was to explore parenting emotion socialization practices and reactions in a clinical sample, to whom parents’ role in the processes related to the child’s emotion regulation can be especially critical.

## 2. Methods

### 2.1. Participants

The sample consisted of 80 primary caregivers (63 mothers, 79%; 10 fathers, 13%; 7 others, e.g., grandmother, 9%) of children referred to a public child and adolescent outpatient psychiatry unit in Portugal. The children were aged 3–13 years (*M* = 8.08, *SD* = 2.63), and the sample mainly included parents of boys (73%). This unit usually receives children and adolescents referred either through the family’s initiative or by schools and other health services, and serves the Lisbon district’s population. Most referrals are related to behavioural problems, inattention, learning difficulties, socialization difficulties, and internalizing problems. A larger group of parents had less than nine years of schooling (41%), and only a minority had a college degree (25%). The mean age was 37.68 years (*SD* = 5.96) for mothers and 38.30 years (*SD* = 4.62) for fathers. Most of the children were living in two-parent families (both parents, 56%; only mother, 22.5%, only father, 2,5%; other family adults, 15%) and had one or more siblings (*M* = 1.60, *SD* = 1.44).

### 2.2. Measures

The Coping with Children’s Negative Emotions Scale (CCNES, [13,26] is a self-reported scale for parents measuring their reactions to children’s negative emotions (e.g., sadness, anger, disappointment, fear). Parents are presented with 12 typical scenarios (e.g., getting nervous about appearing in a recital or sports activity) in which the children experience negative emotions. The parent is asked to indicate the likelihood they would respond in specific ways if that situation occurred with his/her child on a seven-point scale ranging from 1 (very unlikely) to 7 (highly likely). The original subscales reflect six ways parents respond to their children’s negative emotions, specifically: punitive, distress reactions, minimization reactions, problem-focused reactions, expressive encouragement, and emotion-focused reactions. The measure used in this study is a modified version of the Portuguese CCNES [27], where the items of the distress reactions subscale were rephrased to improve clarity [22], and the ignoring subscale was included [26]. Higher scores in a subscale indicate the frequent use of a specific reaction/strategy. In the current sample, the scales’ alpha coefficients were good (Cronbach’s α = 0.83–0.89).

The Child Behavior Check-list (CBCL 1 1/5-5; CBCL 6-18, [28]) measures children’s behaviour and emotional problems. The parent indicates the frequency of each behaviour over the previous six months (CBCL 6-18) or two months (CBCL 1 1/5-5) on a three-point Likert scale ranging from 0 (not true) to 2 (very true or often true). The CBCL 1 1/5-5, composed of 100 items, was used to measure problems for children aged less than six years, and the CBCL 6-18 with 113 items to evaluate problems for children aged six years and above. Higher values indicate higher levels of problems. This study used the total scale and the externalizing and internalizing sub-scales. The alpha coefficients of the externalizing, internalizing subscales, and total scale for the current study sample were good (Cronbach’ α = 0.77–0.93 for CBCL 1 1/5-5 and Cronbach’ α = 0.90–0.95 for CBCL 6-18).

The Lack of Emotional Control/Emotion Dysregulation subscale of the Parent Emotion Regulation Scale (PERS, [22]). This five-item subscale measures parent’s emotional dysregulation and lack of capacity to modulate his/her negative emotionality in order to pursue parenting-related goals. The items are rated on a five-point Likert scale ranging from 0 (never or almost never) to 4 (always or almost always). This subscale showed good reliability for the current sample (Cronbach’ α = 0.72).

The Brief Symptom Inventory (BSI, [29,30]) is a self-reported measure with 53 items covering nine dimensions of parents’ symptoms hostility, anxiety, depression, interpersonal sensitivity, phobic anxiety, somatization, obsession-compulsion, paranoid ideation, and psychoticism). Respondents indicate on a five-point Likert scale ranging from 0 (never) to 4 (very much) the degree to which each problem affected him/her in the previous week. Higher scores indicate higher levels of symptomatology. In this study, only the Global Severity Index was included, which presented a high level of internal consistency for the current sample (Cronbach’ α = 0.96).

Socio-demographic information. Data as to parents’ demographic characteristics (age, highest education level, and marital status), child’s characteristics (age, sex), and family’s characteristics (household composition, number of children, and area of residence) were collected.

### 2.3. Ethical Considerations

Before collecting the data, the study was submitted to the ethics committees of the institutions involved. All participants received an informed consent form explaining the study’s objectives and the voluntary nature of participation.

### 2.4. Data Collection Procedures

Participants were recruited from a public child and adolescent psychiatry outpatient unit serving a diverse population from the Lisbon area. Over one year, all parents of children admitted to this unit were invited by their physician to participate in the study. Of the 242 children admitted, 85 returned the protocol (35%). A significant percentage was not included in the study because the parents did not return the questionnaires or the written informed consent (*n* = 134), the children were living in an institution (*n* = 4), the children were discharged after the first consultation (*n* = 13), or for other reasons (*n* = 3). Only a small percentage refused to participate in the study (*n* = 3).

In total, 85 parents returned the completed questionnaires to the psychiatrist. From these, four cases were eliminated because of missing data (more than three items in CCNES scales), and one was considered invalid because of response bias (i.e., the informant chose the same alternative for all items of the CCNES).

### 2.5. Data Analysis Procedures

Before conducting the main statistical analyses, the following assumptions were tested: for bivariate correlations and model-based clustering, normal distribution; for analysis of variance (ANOVA) normal distribution and homogeneity of variances in each group; for multivariate analysis of variance (MANOVA) normality, homogeneity of variance-covariance matrices (Box M test), linearity, and absence of multicollinearity. Whenever homogeneity of variances was rejected, the *F*-test robustness was evaluated by comparing the ratio of the largest to the smallest variance with the cut-off value of 1.5; a ratio lower than the cut-off suggested the F-test was still robust, and analyses of variance were performed. In addition, data were scanned to identify univariate or multivariate outliers.

First, preliminary analyses were performed to calculate the descriptive statistics of the main study variables and bivariate correlations between variables. A Pearson coefficient was calculated in cases of normally distributed variables. Because minor deviations regarding this assumption were found for some variables, Spearman correlations were estimated for these cases instead.

Second, we cluster analyzed the data related to parents’ reactions to their children’s negative emotions. Due to reasonable normal distribution, a model-based cluster analysis using the mClust package [20] designed for the R environment was performed in this study. The model-based cluster analysis has some advantages in relation to other heuristic cluster algorithms that lack a statistical fit measure to determine the adequacy of the number of clusters and are affected by the sequence of data input or starting values [31]. Specifically, the use of Bayes factors (BIC) to compare models and select clusters negatively impacts the complexity of the model and maximizes parsimonious parameterizations, where the smaller the BIC value, the stronger the evidence for the model [32]. A cluster analysis was performed of the seven parent reactions to children’s emotions: punitive reactions, distress reactions, minimization reactions, ignoring, problem-focused reactions, expressive encouragement, and emotion-focused reactions. The solutions with the best fitting model were compared to retain the more theoretically meaningful one (please, consult the results section for more information on how these decisions were made). To examine the characteristics of parents’ reaction clusters, we conducted a multivariate analysis of variance (MANOVA). We also explored the mean differences between parents’ reactions clusters concerning the child’s age through ANOVA.

Third, four separate ANOVAs were computed to explore the mean differences between parents’ reaction clusters in relation to their dimensions of psychopathological symptoms and emotion dysregulation.

Finally, we explored the differences between parents’ reaction clusters concerning children’s adjustment problems, controlling for the effects of the child’s sex and age. Mean differences between parents’ reaction clusters in relation to children’s externalizing and internalizing problems were explored through MANOVA, followed by two subsequent ANOVAs. We also conducted chi-square independence tests to examine the association between parents’ reaction clusters and clinical status. Clinical status was determined considering the borderline cut-off for each sex and age group in the total CBCL score. Other than clusters analysis, all the analyses were conducted using the SPSS Statistics 26 software package.

## 3. Results

### 3.1. Preliminary Analyses

Only a small proportion of data were missing (<5% for each variable). Missing values were replaced by the mean of the subject for the scale to which the item belonged. No scale had more than three missing items. Table 1 presents the descriptive statistics of and bivariate correlations between the study variables.

Children’s sex and age were not significantly associated with the other variables, apart from a significant positive association of low magnitude between children’s age and parents’ distress reactions (*r* = 0.24). Significant positive associations were evident between parents’ unsupportive reactions and internalizing and externalizing problems. Specifically, higher levels of the four parent unsupportive reactions were associated with higher levels of externalizing problems (*r* = 0.26–0.37), and higher levels of minimization reactions (*r* = 0.30) were associated with higher levels of internalizing problems. No associations were found between positive parent reactions and internalizing and externalizing problems. Additionally, higher levels of parents’ psychopathological symptomatology were positively associated with three of the four unsupportive parent reactions, punitive, distressing, and minimizing reactions (*r* = 0.28–0.40). None of the supportive reactions were significantly associated with parents’ psychopathological symptoms. Finally, parents’ emotional dysregulation had a high-magnitude positive association with unsupportive reactions to children’s emotions (*r* = 0.29–0.45) and a negative association with problem-focused reactions (*r* = 0.22).

### 3.2. Characterizing Parents’ Reactions to Children’s Emotions: Model-Based Cluster Analysis

The results showed that the best fitting models–those with lower BIC values–were a four-cluster solution (different volume and orientation and equal shape) with a BIC of −1450.621 and a three-cluster solution (different volume and orientation and equal shape) with a BIC of −1451.164. These two solutions were compared to ascertain the meaningfulness of each cluster. Although having the lowest BIC value, the fourth cluster solution included two clusters with minimal differences between them. Therefore, for the sake of parsimony, we retained the three-cluster solution. The average certainty classification values for this three-cluster solution (0.95 for cluster 1, 0.96 for cluster 2, and 0.97 for cluster 3) reflected a high degree of classification certainty.

We labelled these clusters *low unsupportive reactions cluster* (Cluster 1, *n* = 21), *high supportive reactions cluster* (Cluster 2, *n* = 40), and *inconsistent reactions cluster* (Cluster 3, *n* = 19). The MANOVA indicated a significant multivariate effect of the cluster factor on the seven parent reaction dimensions (Roy’s Largest Root = 4.91, *F* (7,72) = 50.54, *p* < 0.001, *η*^2^ = 0.99). All of the following ANOVAs demonstrated significant effects of the cluster factor. Table 2 provides the results of the subsequent ANOVAs and the significant findings relative to the mean differences between parents’ clusters and their reactions. Multiple pairwise comparisons using the Tukey-Kramer method for unequal sample sizes were computed to examine how these clusters differed in each dimension of reactions to children’s negative emotions.

The first cluster, the *low unsupportive reactions cluster*, had the lowest levels of unsupportive reactions in the low range and lower values than the *high supportive cluster* of emotion-focused reactions, emotion encouragement, and problem-solving, although all positive reactions were in the moderate range. The second cluster, the *high supportive reactions cluster*, had the highest values of all supportive reactions (emotion encouragement, problem-solving, and emotion-focused reactions) and lower values (compared to the inconsistent cluster) for unsupportive reactions, all in the low range. Finally, the *inconsistent reactions cluster* had the highest values (compared to the two remaining clusters) for punitive, distressed, and minimizing reactions as well as ignoring children’s negative emotions, and lower values for problem-focused and emotion-focused reactions (compared with the high supportive reactions cluster). In this cluster, all supportive and unsupportive reactions were in the moderate range. Results of the ANOVA (*F* (2,80) = 1.332, *p* = 0.270) showed no significant statistical differences between the clusters in relation to the child’s age (Table 3).

### 3.3. Parents’ Clusters and Parents’ Psychopathological Symptoms and Emotion Dysregulation

We conducted ANOVAs (Table 3) to investigate whether parents’ clusters were associated with parents’ psychopathology and emotion dysregulation. Multiple pairwise comparisons using Tamhane (equal variances not assumed for the dimension “orientation to children’s emotions”) and Tukey-Kramer (equal variances assumed for all other variables with unequal sample sizes) tests were performed to examine which clusters differed from each other. The results showed that parents of the *inconsistent reactions* cluster reported the highest levels of parent emotion dysregulation, and those of the *low unsupportive reactions cluster* showed the lowest levels of parent psychopathology.

### 3.4. Parents’ Clusters and Children’s Externalizing and Internalizing Problems

The MANOVA showed a significant multivariate effect (Roy’s Largest Root = 0.19, *F* (2,73) = 6.75, *p* < 0.001, *η*^2^ = 0.16) of the cluster factor on children’s internalizing and externalizing problems, controlling for the effects of child age and sex. The two subsequent ANOVAs also showed the significant effects of the cluster factor (Table 4).

Multiple pairwise comparisons using Bonferroni correction were used to examine which clusters differed from each other. The *inconsistent* cluster showed the highest levels of externalizing and internalizing problems (differed significantly from the other two clusters).

Finally, a chi-square test of independence indicated significant associations between the clusters and children’s clinical status (χ^2^(2) = 8.77, *p* = 0.012, Cramer’s *V* = 0.34) according to the borderline clinical values for the total CBCL (Table 5). The inconsistent cluster showed more children above the borderline clinical level than expected.

## 4. Discussion

Parents play a central role in children’s emotional development [3,7]. A critical mechanism of emotion socialization is how parents respond to children’s emotions [3,6]. Gottman and colleagues [6] proposed that parents consistently respond to their children either with emotion coaching, valuing emotions, and supporting emotion-related practices, or dismissing emotions and using unsupportive emotion-related practices. The current study adopted a person-centred approach to characterize parental reactions to children’s negative emotions in a clinical sample. We explored whether these profiles were characterized by parents’ consistent responses and how these different patterns related to the child and the parent’s adjustment. The results supported consistent and inconsistent patterns and showed that these patterns were significantly associated with the children’s and the parents’ adjustment.

Three profiles of parents’ emotion reactions emerged in the study: *low unsupportive*, *high supportive, and inconsistent reaction patterns.* The *low unsupportive reactions cluster* (*n* = 18) showed the lowest levels of unsupportive and supportive reactions, although the supportive reactions were moderate. This seems to represent a group of parents with lower (positive or negative) reactivity to children’s negative emotions, which can be explained by parents’ characteristics (e.g., temperament, mental health, emotion regulation capabilities). This cluster also presented the lowest levels of parents’ psychopathology and lower levels of parents’ emotional dysregulation. Having adequate mental health can help these parents to better regulate their emotions in the parenting context and, consequently, manifest fewer intensive reactions to their child’s negative emotions [33].

The second, *high supportive cluster* (*n* = 13), showed low unsupportive reactions and the highest supportive reactions to children’s emotions. This cluster is the most similar to the emotion coaching philosophy. It includes three of the five components identified by the authors as characteristics of emotion coaching philosophy: parents seem to validate their child’s emotion, encourage their child to express their emotions, and problem-solve with their child, helping them manage the situation that led to the negative emotion. Although we did not analyze more global parental attitudes or beliefs toward emotions, it is expected that these parents also value the role of emotions in the child’s development and are aware and oriented towards their child’s emotions. On the other hand, these parents also manifested less emotional dysregulation than parents in the inconsistent reaction pattern. To be a responsive and emotion coaching parent in high emotional arousal situations requires high competence in regulating their own emotions [20].

Finally, the third cluster, *inconsistent reactions*, was characterized simultaneously by moderate levels of unsupportive and supportive reactions to children’s negative emotions. This finding seems to replicate previous research using a person-based approach that found reaction patterns characterized by inconsistent responding [18]. This cluster also demonstrated the highest levels of parents’ psychopathology and emotional dysregulation. A previous study [13] examined day-to-day consistency of parental reactions to children’s emotions in a high-risk group of substance-dependent mothers and found significantly higher supportive and non-supportive reactions to children’s negative emotions compared to a non-clinical group. They also found that these mothers engaged in significantly higher levels of inconsistent reactions during periods of problematic drug use than periods of sobriety.

Parents’ mental health problems and emotional dysregulation can prevent them from taking a positive approach to understanding their children’s emotions in difficult situations [33]. Consistent with Dix’s [34] model, emotions are needed to motivate parents to engage with and effectively respond to the child; however, strong negative emotions can interfere with parents’ resources (e.g., attentional resources, problem-solving capacities), which are critical in sensitive and responsive parenting. Therefore, parents’ sensitive responses depend on their ability to modulate their own negative emotions to accomplish their goals related to parenting.

Similarly, parents’ ability to regulate their own emotions can be important when working with parents of children with emotional and behavioural problems who have more frequent and intense emotional displays. These parents need to respond (and not impulsively react) to the intense emotional reactivity of their children, focusing on the needs of the child and responding accordingly, sometimes helping the child to understand and modulate their emotions, at other times setting limits to his/her behaviour, and in other occasions helping them to solve the problem that originated the negative emotion. This flexibility depends on the parent’s capacity to be attentive to the child’s needs, requiring an adequate regulation of the parent’s emotional states.

The three clusters also differed as to the children’s internalizing and externalizing problems. The *inconsistent reactions* cluster had the worst results regarding the child’s psychopathology, consistent with previous empirical work showing a significant association between inconsistent parents’ emotion socialization and poorer child outcomes [12]. A transactional perspective of parents’ behaviour can also support these findings, with parents’ unsupportive reactions to children’s emotions contributing to increased adjustment difficulties and children’s extreme reactions and dysregulation provoking parents’ ambivalent strategies.

The ambivalent responses that characterize the *inconsistent reactions* cluster resemble Granic and Patterson’s [35] description of the parent-child dyadic pattern of families with aggressive children, marked by simultaneously hostile and permissive parenting behaviour. This parent–child dyadic pattern involves both behaviours and affective-cognitive processes interacting with each other in feedback cycles repeated over time, progressively becoming more stable. In the *inconsistent*
*reactions* cluster, the parent can start by responding positively to the child’s negative emotions, but when the parent perceives an increase in emotional pressure with the child’s negative emotions and behaviour escalating, they revert to punitive and unsupportive strategies. In turn, these strategies increase the likelihood of the child reciprocating with negative emotions/behaviour, amplifying his/her initial negative emotional display. This interaction pattern may be facilitated by the parent’s lack of emotional control when interacting with the child.

On the other hand, ambivalence can result from the parents’ responding differently to different situations, sometimes being more supportive and other times more punitive. This inconsistent responding may also reveal various parental attributions regarding their child’s behaviour in different situations, sometimes understanding the negative emotional display in the context of the child’s emotional regulation difficulties, and in other instances perceiving the same behaviour as intentional and controllable by the child.

The measures used in this study do not allow us to fully understand the meaning of this inconsistency. In the future, it would be important to further explore the meaning of parents’ inconsistent responses to the test if they imply sequentially ambivalent responses as in the Granic and Patterson [29] model or if they indicate other processes.

Finally, although a developmental analysis was not the focus of this study, the associations between parents’ reactions clusters and children’s ages were explored. Throughout development, children develop emotion regulation capacities and become more independent from their parents through this process, and parents are consequently expected to modify the way they support their children in managing emotional situations. The lack of significant associations between the children’s age and parents’ clusters is not in accordance with this literature that suggests significant changes in emotion socialization strategies over time. The fact that this is a clinical sample, including children with more emotional regulation problems and who may depend more on adults to help them regulate these emotions (even the older ones), may explain the lack of significant associations. Furthermore, the small sample size of the different age groups in this study (early childhood, middle childhood, and pre-adolescence) also limited a developmental analysis. In the future, it would be interesting to explore if the emerging clusters differ for different age groups and if age moderates the relation between parenting emotion socialization practices and the child’s adjustment.

In conclusion, these results indicate that most parents present consistent responses to their child’s negative emotionality, reinforcing previous empirical studies’ results using a variable-centred approach (e.g., [12]). This result is theoretically consistent with the concept of emotion socialization practices (including parents’ responses to children’s emotions) as active, intentional, and purposive behaviours that parents endorse to communicate their values and goals regarding emotions [3]. However, a group of parents can demonstrate different responses in dealing with their child’s negative emotions. These results are aligned with previous studies that adopted a person-based approach [16,17]. This lack of parental consistency may be due to several factors, including parents’ emotional regulation competencies and children’s mental health and temperamental characteristics. The higher levels of parents’ emotion dysregulation and children’s adjustment problems observed in the inconsistent pattern support the idea that parents’ consistency might be a greater challenge under these conditions.

Several limitations of this study should be considered. First, this was a cross-sectional study, and it therefore did not allow conclusions as to the directions of effects. Longitudinal studies could determine how the transactions between parents’ reactions to their child’s emotions and adjustment unfold during the child’s development. Second, the reliance on only one informant can increase the risk of common method variance. Besides, relying only on parents’ reports to evaluate their reactions to their child’s emotions is a significant limitation, as the parents may have limitations in recognizing their children’s emotions and their own responses, and their responses may be subject to desirability bias. Future studies should consider including other methodologies (e.g., observation of parent-child interactions in situations eliciting emotion) and informants (e.g., children’s self-reports of their own emotional and behavioural problems) to evaluate parents and children variables. Third, most respondents were mothers, and data from mothers and fathers were not independently analyzed. Some research suggests important interactional effects of mothers’ and fathers’ reactions to children’s emotions on the child’s adjustment [36]. It would be interesting to cluster analyze the reactions of mothers and fathers to investigate these interactional effects. Fourth, the measure to evaluate parents’ reactions to their child’s negative emotions was originally created to evaluate these reactions among parents of younger children (pre-school or elementary school). While the scale did demonstrate adequate reliability in an older sub-sample of the current study, some parents’ responses to their children’s negative emotions may change across the development stages, which this measure may not capture. Fifth, the limited sample size prevented more sophisticated analyses and exploring the effects of parents’ reactions clusters on children’s emotional adjustment controlling for other critical dimensions (e.g., child’s sex, parents’ education and sex.). In the future, it would also be interesting to analyze the role of the child’s age as a moderator in the relationship between parents’ reactions to children’s emotions and children’s adjustment. Finally, the sample included mainly parents with low formal education. Nevertheless, this sample seems representative of the adult Portuguese population, given that only 26% of Portuguese adults have a college degree.

The fact that this sample includes more diversity in terms of socioeconomic background and that this study was conducted in Portugal can be regarded as relevant since most of the research on parental emotion socialization was conducted with North American and middle-class families [7,37]. The patterns of parental reactions found–consistent and inconsistent–and their links to parents’ psychopathological symptoms and emotion dysregulation and to children’s adjustment were in accordance with previous studies and expand the literature on parents’ socialization of emotion. This supports the idea that the response categories of parent’s reactions to a child’s negative emotions may have relevance across cultures [37] and that the relation between these and child and parent characteristics may not differ substantially [7], at least in Western countries.

## 5. Conclusions

To our knowledge, this is the first study to adopt a typological approach to investigate parents’ reactions to children’s emotions in a clinical sample. The results of this study suggest the importance of supporting parents so they can properly manage their emotions and be more sensitive in their response to their children’s emotional demonstrations. Most preventive and therapeutic evidence-based interventions are grounded in behavioural models and centred on behavioural modification strategies [38]. More recently, interventions centred on promoting parents’ support to children’s self-regulation and parents’ emotion regulation were developed (e.g., “Tuning in to Kids” [39]), with promising results [33].

This support is even more critical when intervening with the parents of children with behavioural and emotional disorders. When intervening with children with externalizing and internalizing problems, it is necessary to work with parents so that they understand the role of emotions in children’s development, respond in a more sensitive way to negative emotional displays, and help the child understand and modulate their own emotions. This study also suggests that parents with higher levels of emotional dysregulation may have more problems maintaining a consistent pattern of responding. Screening for this kind of behaviour and supporting parents in regulating their own emotions can be relevant in these situations.

The advantages of including a parental emotion regulation component when working with parents of children with mental health problems are not sufficiently studied. Some preliminary research suggests that adding this component enhances the results of traditional parenting programs [40,41,42], while other studies point in the opposite direction [43]. Future studies should further explore the role of parents’ emotion socialization in clinical samples and evaluate the impact of interventions directed to increase parents’ support to children’s emotion regulation and parent’s emotion regulation.

## Figures and Tables

**Table 1 ijerph-19-06844-t001:** Descriptive statistics and bivariate correlations between the study variables.

	2	3	4	5	6	7	8	9	10	11	12	13	*M*	*SD*
1. Child Sex	0.17	0.09	0.10	−0.02	0.14	0.18	0.04	0.06	−0.10	0.01	−0.07	0.01		
2. Child Age		0.06	0.24 *	0.05	0.02	0.05	0.14	−0.03	−0.20	0.22	−0.01	0.09	8.08	2.63
3. CCNES_PR	-	-	0.76 ***	0.79 ***	0.68 ***	0.01	0.09	−0.01	0.37 ***	0.20	0.45 ***	0.40 ***	2.46	1.13
4. CCNES_DR	-	-	-	0.67 ***	0.66 ***	−0.05	0.13	−0.05	0.23 *	0.22	0.44 ***	0.40 **	2.43	1.26
5. CCNES_MR	-	-	-	-	0.57 ***	0.18	0.23 *	0.15	0.26 *	0.30 **	0.34 **	0.29 **	3.48	1.10
6. CCNES_I	-	-	-	-	-	−0.12	0.13	−0.02	0.21	0.02	0.29 *	0.28 *	2.04	1.06
7. CCNES_PF	-	-	-	-	-	-	0.62 ***	0.79 ***	−0.17	0.02	−0.22 *	−0.08	5.39	1.01
8. CCNES_EE	-	-	-	-	-	-	-	0.58 ***	−0.15	0.08	−0.08	−0.03	4.77	1.19
9. CCNES_EF	-	-	-	-	-	-	-	-	−0.07	0.04	−0.17	−0.03	5.4	0.98
10. CBCL_Ext	-	-	-	-	-	-	-	-	-	0.43 ***	0.22 *	0.29 *	14.90	9.55
11. CBCL_Inter	-	-	-	-	-	-	-	-	-	-	0.09	0.56 ***	15.38	8.61
12. PERS_EmotDysr	-	-	-	-	-	-	-	-	-	-	-	0.39 ***	1.18	0.66
13. BSI GSI	-	-	-	-	-	-	-	-	-	-		-	1.66	0.45

Note. CCNES: Punitive Reactions (CCNES_PR), Distress Reactions (CCNES_DR), Minimization Reactions (CCCES_MR), Ignoring (CCNES_I), Problem-Focused Reactions (CCNES_PF), Expressive Encouragement (CCNES_EE), Emotion-focused Reactions (CCNES_EF); CBCL: Externalizing Problems (CBCL_Ext); Internalizing Problems (CBCL_Int); BSI: Global Severity Index (BSI_GSI); Parents’ Emotion Dysregulation/Lack of Emotional Control (PERS_EmotDysr), * *p* ≤ 0.05. ** *p* ≤ 0.005. *** *p* ≤ 0.001.

**Table 2 ijerph-19-06844-t002:** Comparisons between parents clusters: means (*M*), standard deviations (*SD*), ANOVAs results, and multiple pairwise comparisons.

	Cluster 1 Low Unsupportive (*n* = 21)	Cluster 2 High Supportive (*n* = 44)	Cluster 3 Inconsistent (*n* = 15)	*F*	*η* ^2^	Multiple Pairwise Comparisons
	*M* (*SD*)	*M* (*SD*)	*M* (*SD*)			
CCNES_PR	1.67 (0.56)	2.24 (0.76)	4.19 (0.84)	*F* (2,77) = 56.00	0.59	I > HS > LR
CCNES_DR	1.30 (0.28)	2.38 (0.65)	4.18 (0,75)	*F* (2,77) = 101.69	0.73	I > HS > LR
CCCES_MR	2.80 (0.99)	3.32 (0.73)	4.92 (0.90)	*F* (2,77) = 30.15	0.44	I > HS, LR
CCNES_I	1.19 (0.25)	1.99 (0.74)	3.34 (1.23)	*F* (2,77) = 35.10	0.48	I > HS > LR
CCNES_PF	4.96 (1.28)	5.74 (0.72)	4.96 (0.90)	*F* (2,77) = 6.67	0.15	HS > LR, I
CCNES_EE	3.81 (1.39)	5.19 (0.78)	4.88 (1.19)	*F* (2,77) = 12.48	0.25	HS > LR
CCNES_EF	4.80 (1.18)	5.83 (0.66)	5.40 (0.98)	*F* (2,77) = 12.15	0.24	HS > I, LR

Note. CCNES: Punitive Reactions (CCNES_PR), Distress Reactions (CCNES_DR), Minimization Reactions (CCCES_MR), Ignoring (CCNES_I), Problem-Focused Reactions (CCNES_PF), Expressive Encouragement (CCNES_EE), Emotion-focused Reactions (CCNES_EF).

**Table 3 ijerph-19-06844-t003:** Comparisons between parents’ clusters in relation to parents’ emotion dysregulation, psychopathological symptoms, and child’s age: means (*M*), standard deviations (*SD*), ANOVAs results, and multiple pairwise comparisons.

	Cluster 1 Low Unsupportive (LR) (*n* = 21)	Cluster 2 High Supportive (HS) (*n* = 44)	Cluster 3 Inconsistent (I) (*n* = 15)	*F*	*η* ^2^	Multiple Pairwise Comparisons
	*M* (*SD*)	*M* (*SD*)	*M* (*SD*)			
Child’s age	7.43 (2.46)	8.11 (2.60)	8.87 (2.88)	*F* (2, 77) = 1.33	0.03	n.s.
PERS_EmotDysr	0.96 (0.66)	1.14 (0.63)	1.62 (0.58)	*F* (2,79) = 4.98 *	0.12	I > HS, LR
BSI GSI	1.38 (0.27)	1.74 (0.47)	1.83 (0.46)	*F* (2,77) = 6.41 **	0.15	I, HS > LR

Note. Parents’ Emotion Dysregulation/Lack of Emotional Control (PERS_EmotDysr); BSI: Global Severity Index (BSI_GSI);.* *p* ≤ 0.05. ** *p* ≤ 0.005.

**Table 4 ijerph-19-06844-t004:** Comparisons between clusters in relation to children’s internalizing and externalizing problems: means (*M*), standard deviations (*SD*), ANOVAs results, and multiple pairwise comparisons.

	Cluster 1 Low Unsupportive (LR) (*n* = 21)	Cluster 2 High Supportive (HS) (*n* = 44)	Cluster 3 Inconsistent (I) (*n* = 13)	*F*	*η* ^2^	Multiple Pairwise Comparisons
	*M* (*SD*)	*M* (*SD*)	*M* (*SD*)			
CBCL_Intern	13.29 (5.30)	14.52 (9.14)	21.69 (8.73)	*F* (2,73) = 3.98 *	0.10	I> LR, HS
CBCL_Extern	13.62 (9.49)	13.41 (8.97)	22.00 (9.00)	*F* (2,73) = 5.12 *	0.12	I> LR, HS

Note. CBCL: Externalizing Problems (CBCL_Ext); Internalizing Problems (CBCL_Int) * *p* ≤ 0.05.

**Table 5 ijerph-19-06844-t005:** Distribution of children above the clinical borderline level between parents’ reactions clusters: Absolute, relative frequencies, and adjusted residuals.

	Cluster 1 Low Unsupportive (*n* = 21)	Cluster 2 High Supportive (*n* = 44)	Cluster 3 Inconsistent (*n* = 13) ^a^
	*n* (%)	*n* (%)	*n* (%)
Total CBCL above the clinical borderline	11 (52.4)	27 (61.4)	13 (100)
Adjusted residual	−1.5	−0.8	2.9

Note. CBCL Total score: CBCL_Total. ^a^ Two missing values.

## Data Availability

Data are available from https://osf.io/jbvxg/?view_only=603148aaf2344b1ebeb54b11884beb45 (accessed on 2 June 2022).

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
