# Peer review of "Patterns of Parental Reactions to Their Children’s Negative Emotions: A Cluster Analysis with a Clinical Sample"

_ijerph, 2022, doi:10.3390/ijerph19116844_

Round 1
Reviewer 1 Report
This is a study that makes significant contributions to the understanding of parental reactions to the child's negative emotions. The study is complex and rigorously conducted. The rigour of the methodology adds value, of course. Method: I would only recommend that the program with which the statistical analyzes were performed be reproduced.
I would like to ask the authors if the CCNES is validated for Portuguese. If not, I suggest entering some information about adapting the scale for this study. Ex: who did the translation? Responses to CCNES may be biased by social desirability. How is this aspect treated in the study?
The Brief Symptom Inventory - missing reference in parentheses
Page 9 - text description of the data in table 2. I recommend that the correlation coefficients r be inserted in parentheses.
Author Response
This is a study that makes significant contributions to the understanding of parental reactions to the child's negative emotions. The study is complex and rigorously conducted. The rigour of the methodology adds value, of course.
We thank the reviewer for his/her positive appreciation of this work.
Method: I would only recommend that the program with which the statistical analyzes were performed be reproduced.
We thank the reviewer for this suggestion. Cluster analysis was performed through the mClust package [20] designed for the R environment (p.11, lines 3/4). All the other procedures were conducted using the SPSS Statistics 26 software package (p.12, lines 4/5).
I would like to ask the authors if the CCNES is validated for Portuguese. If not, I suggest entering some information about adapting the scale for this study. Ex: who did the translation?
The Portuguese version of CCNES was originally developed and studied by Alves and Cruz (2011). The current version presents a modified version of this scale where the items of the distress reactions subscale were rephrased to improve clarity (Pereira et al., 2017) and to include the ignoring subscale (Mirabile, 2015). This information was added in the respective section (p.8, 1st paragraph).
Alves D & Cruz O Reacções parentais às emoções negativas dos filhos (RPEN): um questionário de avaliação da metaemoção parental [Parental reactions to children’s negative emotions: A questionnaire to evaluate parental meta-emotion] . In S. Ferreira, A. Verhaeghe, D. R. Silva, L. S. Almeida, R. Lima & S. Fraga (Eds.), Actas do VIII congresso iberoamericano de avaliação/evaluación psicológica e XV conferência internacional avaliação psicológica: formas e contextos. 2011; 1480-92, Lisboa, SPP.
Mirabile SP. Ignoring Children's Emotions: A novel ignoring subscale for the Coping with Children's Negative Emotions Scale. European Journal of Developmental Psychology. 2015;12(4):459-71
Pereira AI, Barros L, Roberto MS, Marques T. Development of the Parent Emotion Regulation Scale (PERS): Factor structure and psychometric qualities. Journal of Child and Family Studies. 2017; 26(12), 3327–38.
Responses to CCNES may be biased by social desirability. How is this aspect treated in the study?
Yes, that is a possibility, since it is a self-report measure. We included this shortcoming in the discussion (p.20/21).
The Brief Symptom Inventory - missing reference in parentheses
Thank you very much for noticing that. The references were included:
Canavarro C. Inventário de Sintomas Psicopatológicos: BSI [Inventory of psychopathological symptoms]. In: Gonçalves M, Almeida LS, editors. Testes e provas psicológicas em Portugal II. Braga: SHO/APPORT; 1999. p. 87-109.Derogatis LR, Melisaratos N. The Brief Symptom Inventory: an introductory report. Psychol Med. 1983;13(3):595-605.
Page 9 - text description of the data in table 2. I recommend that the correlation coefficients r be inserted in parentheses.
We think the reviewer is mentioning the description of the data in Table 1. We inserted the coefficients in the text, as suggested (p.12/13).
We thank the reviewer for his/her careful reading of the manuscript and his/her constructive remarks and suggestions.
Reviewer 2 Report
The articcle presents an original paper. For futre researchs it would be neccessary to level the sample and expand it.
More attention should be paid to the different sociocultural variables.
Author Response
The article presents an original paper. For future research it would be necessary to level the sample and expand it.
More attention should be paid to the different sociocultural variables.
Thank you very much to the reviewer for this suggestion. We added a paragraph in the discussion section relative to sociocultural variables:
“The fact that this sample includes more diversity in terms of socioeconomic background and that this study was conducted in Portugal can be regarded as relevant since most of the research on parental emotion socialization was conducted with North American and middle-class families (Eisenberg, 2020; Raval & Walker, 2019). The patterns of parental reactions found, consistent and inconsistent, as well as their links to parents’ psychopathological symptoms and emotion dysregulation, and children’s adjustment, were in accordance with previous studies and expand the literature on parents’ socialization of emotion. This supports the idea that the response categories of parent’s reactions to a child’s negative emotions may have relevance across cultures (Raval & Walker, 2019) and that the relation between these and child and parent characteristics may not differ substantially (Eisenberg, 2020) , at least in western countries.
Eisenberg N. Findings, issues, and new directions for research on emotion socialization. Dev Psychol. 2020;56(3):664-670. doi:10.1037/dev0000906
Raval V.V., Walker B.L. Unpacking ‘culture’: Caregiver socialization of emotion and child functioning in diverse families. Dev Rev. 2019; 51, 146–174.
Reviewer 3 Report
Please add in the methods:
Ethical considerations sutitle in which you write all details of the participant's consent.
also there are too many old references,please update references
Author Response
Please add in the methods:
Ethical considerations sutitle in which you write all details of the participant's consent.
As suggested by the reviewer, we inserted a subsection of ethical considerations (p.9):
Ethical Considerations
Before collecting the data, the study was submitted to the ethics committees of the institutions involved. All participants received an informed consent form explaining the study's objectives and the voluntary nature of participation.
also there are too many old references,please update references
Thank you for the suggestion. Old references were maintained because they represent seminal contributions to the parents’ emotional socialization literature, but we included some new and updated references. For example:
Eisenberg N. Findings, issues, and new directions for research on emotion socialization. Dev Psychol. 2020;56(3):664-670. doi:10.1037/dev0000906
Hajal NJ, Paley B. Parental emotion and emotion regulation: A critical target of study for research and intervention to promote child emotion socialization. Dev Psychol. 2020;56(3):403-417. doi:10.1037/dev0000864
Raval V.V., Walker B.L. Unpacking ‘culture’: Caregiver socialization of emotion and child functioning in diverse families. Dev Rev. 2019; 51, 146–174.